# CLASP: Constrained Latent Shape Projection for Refining Object Shape from Robot Contact

**Brad Saund and Dmitry Berenson**
Robotics Institute
University of Michigan
bsaund@umich.edu, dmitryb@umich.edu

**Abstract:** Robots need both visual and contact sensing to effectively estimate the state of their environment. Camera RGBD data provides rich information of the objects surrounding the robot, and shape priors can help correct noise and fill in gaps and occluded regions. However, when the robot senses unexpected contact, the estimate should be updated to explain the contact. To address this need, we propose CLASP: Constrained Latent Shape Projection. This approach consists of a shape completion network that generates a prior from RGBD data and a procedure to generate shapes consistent with both the network prior and robot contact observations. We find CLASP consistently decreases the Chamfer Distance between the predicted and ground truth scenes, while other approaches do not benefit from contact information.

## 1 Introduction

You look into a cabinet and see a box of crackers. You reach in and attempt to grab the box from the side, but your fingers hit something. Perhaps this box is larger than you thought? Your mental model of the box updates, you try a wider grasp, and you successfully retrieve your snack. Robots are currently not so adept. While they can estimate the pose of known shapes [1] or estimate parameters of objects [2], they cannot yet fuse this visual and contact information to draw from the wide range of shape priors in the world. A robot could try to learn its next action directly from vision and force feedback instead [3], but this approach lacks the logic to generalize to scenarios not seen in training.

We propose a method that allows robots to mimic the process of updating object shape from contact information. A shape completion neural network first generates beliefs over possible object shapes based on visual RGBD data. The belief updates the object shape to be consistent with contact information gathered by a robot moving in the scene. We make the realistic assumption that the RGBD camera perceiving the scene suffers from sensor noise and occlusion. We assume the robot can sense *if* it collides with an object, but not *where* the contact was made (i.e., no sensorized skin). Many of the "cobot" platforms available today utilize this contact model to detect collision and stop before harming a person.

Formally, this type of contact creates a contact manifold, a thin space of shapes with a boundary bordering the robot. Past work has projected shapes onto the contact manifold in object pose space [4] and robot configuration space [1], but both require known shape geometry. Our objective is to update the unknown shape geometry to satisfy contact constraints. Returning to the cracker box example, the robot will be unsure if the contact occurred at the top finger, the bottom finger, or perhaps the back of the hand or the elbow. Filling in all possible contact points would lead to absurd scenes with robot shells protruding from the cracker box. However, ignoring this contact information leaves the robot with the original belief of the thinner cracker box and no explanation of why the attempted grasp failed. Our shape completion network generates a prior in latent shape space which can be decoded into shapes in workspace; however, shapes generated directly from this latent prior are unlikely to satisfy contact constraints.

Code is available at https://github.com/UM-ARM-Lab/contact_shape_completion

5th Conference on Robot Learning (CoRL 2021), London, UK.

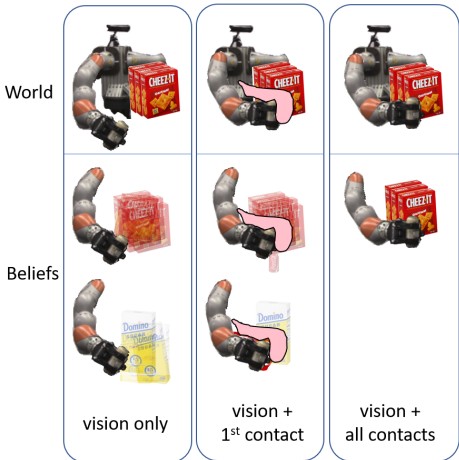

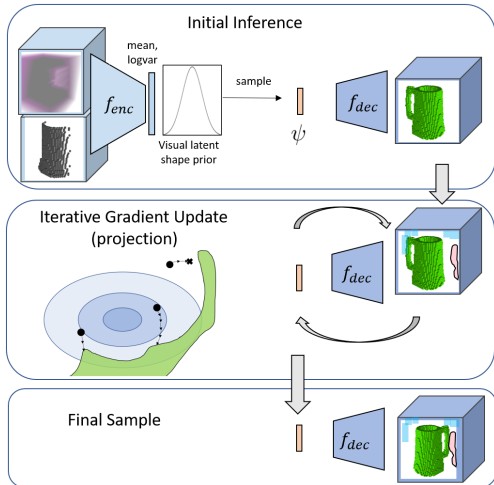

Figure 1: A visual RGBD view of objects leave ambiguity final shape due to sensor noise and occlusion, which we store as a set of sampled scenes in a particle filter. Contact information (pink) reduces ambiguity, and using CLASP the particles converge to the true shape.

Figure 2: CLASP Architecture
**Top:** Shapes sampled from RGBD using PSSNet [5].
**Middle Right:** A robot motion detects free space (light blue) and a collision set (pink).
**Middle Left:** Latent samples are projected to satisfy contacts (green). Ovals depict the latent prior.
**Bottom:** Final samples satisfy the contact constraints.

Our *key insight* is that latent samples from our neural network can be projected onto the contact manifold in the latent shape space using iterative gradient descent, creating shapes both likely under the visual prior and consistent with the contact information. We further expect these projected shapes to be closer to ground truth than direct samples not considering contact.

We accomplish this with our proposed Constrained LAtent Shape Projection (CLASP), which stores a belief over shapes in a particle filter. Each particle represents a collection of latent object shapes which can be decoded into a scene. Every new robot measurement of contact and freespace triggers an update on all particles. During each particle update, gradient steps are taken to increase the occupancy likelihood of the most likely point(s) explaining the contact(s), decrease the occupancy likelihood of the free points, and increase the latent likelihood under the shape prior.

We test this method both in simulation and on a live robot by constructing scenes of objects on a tabletop, generating robot motions that generate freespace and contact observations (Fig. 1), updating the belief using CLASP as well as baselines, and comparing the set of sampled shapes to the ground truth scene. We find CLASP outperforms both ablations of CLASP, as well as the approaches of Rejection Sampling and directly updating the input to the shape completion network. We also find that CLASP produces more accurate scenes than a VAE_GAN shape completion network.

## 2 Related Work

**Shape Completion from Vision:** The goal of Shape Completion is to predict a full shape from a single partial input. Recently, neural networks have become a popular method of shape completion. A common network architecture learns an encoder to a feature space followed by a decoder to the shape output [6, 7, 8, 9, 10, 11, 12, 13, 14, 15, 16, 17]. Scene Completion networks are trained on larger spatial volumes of occupied points and use similar architectures with adaptations to join information from multiple scales [18]. While most methods predict the single best estimated shape, we build off work that uses a variational autoencoder architecture to produce plausible and diverse shapes [5]. We draw from the vast work on shape and scene completion and contribute a method that improves scene estimates using contact information from a robot.

**Shape Completion from Touch:** In different works, "touch" can refer to a single known contact point, a contact configuration, force-torque measurements, or a rich tactile sensor. Work using the

definition of contact point or contact configuration typically uses touch to reduce the version space of shape possibilities [19], but such approaches cannot tractably capture the diversity of all shapes. Alternatively, some situations model known shapes with unknown poses [2]. Both classical Iterative Closest Point [20, 21] and neural networks [22, 23, 24] have been used to predict valid poses. To generate samples consistent with contact information, the Implicit Manifold Particle Filter projects sampled poses onto the contact manifold using an iterative approach [4]. Analogously, we project sampled particles onto the contact manifold in the latent shape space of our neural network.

Touch can also refer to the rich tactile sensors such as GelSight [25] or soft-bubble grippers [26], with input more analogous to images. Tactile patches can be directly mapped to visual features [27]. Alternatively, neural networks have used these sensors for material classification [28] and grasped pose estimation [29]. We do not assume our contact sensing has such rich information.

**Combining Vision and Touch:** A neural network can combine vision and touch (force + torque [3], or GelSight [30]) using separate encoders to a latent space for each sensing modality alongside a decoder to a variety of spaces. We considered a similar encoder structure with a decoder to produce completed shapes, but this would require a large dataset of (Shapes × Contact + Freespace) measurements and the resulting network would be only applicable to the robot used for training. The loss of a neural network can be tweaked during training to bias towards priors similar to contact, such as connectivity and stability [31]. A neural network can output a downstream objective, such as grasp success probability, instead of shape reconstruction, and thus may be successful for a diverse array of objects where accurate reconstruction is not available [32]. Insteaed of a neural network, surface reconstruction has be done using a Gaussian Proccess (GP) prior fit to tactile measurements [33, 34].

Our method is most similar to the work of Wang et al. [35], which uses gradient descent on the latent space of a shape completion network to enforce touch constraints. Where that work uses a high-resolution GelSight tactile sensor to refine shape details previously reconstructed from vision, our work focuses on ambiguous shapes (e.g. a box with unknown depth, or novel shapes not in the training data) and the lower information measurement of contact detection. We accomplish much larger shape updates by using a diverse set of predictions and a novel projection loss function.

## 3  Problem Formulation

Consider a robot $\mathcal{R}$ observing a static scene composed of specific objects $o_j$ sampled from some distribution of objects $\mathcal{O}$. The objects divide the workspace into occupied space $\mathcal{W}_{occ}$ and free space $\mathcal{W}_{free} = \mathcal{W} \setminus \mathcal{W}_{occ}$ The robot has access to a training subset of $\mathcal{O}$ beforehand, but does not know the specific objects $o_j$ in the current scene.

The robot observes the scene with two distinct sensing modalities. In the visual modality, the robot views the scene from a stationary RGBD camera receiving color depth images $Im$. Due to sensor noise and occlusion these depth images offer an incomplete and noisy measurement on the full region of $\mathcal{W}_{occ}$ occupied by the obstacles. From the camera image we assume the scene can be segmented into distinct objects from $\mathcal{O}$.

For the tactile modality, consider a robot that is able to sense *if* it has made contact with any object, but not *where* along the robot surface the contact was made. We assume the contact does not move the objects. For a configuration in configuration space $q \in \mathcal{C}$, let $\mathcal{R}(q) \subset \mathcal{W}$ denote the region of workspace occupied by the robot. A robot that has visited configurations $\{q_1, q_2, ...\} = Q_{free} \subset \mathcal{C}$ without observing contact can carve out regions of known free space:

$$\bigcup_{q \in Q_{free}} \mathcal{R}(q) = \mathcal{W}_{known\_free} \subset \mathcal{W}_{free} \tag{1}$$

For each configuration $q_{contact} \in Q_{contact}$ where contact is observed, there must be at least one object point in collision with the robot (and not in known freespace).

$$\forall q_{contact} \in Q_{contact} \, \exists \, p_{contact} \in \big(\mathcal{R}(q_{contact}) \setminus \mathcal{W}_{known\_free}\big) : p_{contact} \in \mathcal{W}_{occ} \tag{2}$$

Using existing nomenclature, each such region is called a Collision Hypothesis Set (CHS) [36].

Our objective is to model the conditional occupancy $p(\mathcal{W}_{occ}|\mathcal{O}, Im, Q_{free}, Q_{contact})$. Specifically, we desire a stochastic function $g(\mathcal{O}, Im, Q_{free}, Q_{contact})$ which generates sample $\mathcal{W}_{occ}$ as similar as possible to the true conditional distribution. Since the true conditional distribution is unknown, in practice we seek to minimize the distance of drawn samples to the ground truth scene.

# 4 Method

Our approach is to use a particle filter storing a collection of latent shapes. We first segment the scene into distinct objects, then use an existing shape completion neural network to draw latent shape samples $\psi_j$ for each object from $p(\psi_j | \mathcal{O}, Im)$, initializing the particle filter. Each particle can be decoded into the objects in a scene, thus the collection of particles represents the belief $p(\mathcal{W}_{occ} | \mathcal{O}, Im)$. We propose Constrained LAtent Shape Projection (CLASP) as the measurement update, projecting these samples onto the constraints imposed by $Q_{free}$ and $Q_{contact}$. Our method is shown in Fig. 2, where the trapezoids $f_{enc}$ and $f_{dec}$ are the encoders and decoders of PSSNet [5].

## 4.1 Initial Belief

The RGBD camera images are passed to a segmentation algorithm, which yields distinct pixel regions in the image corresponding to different objects $o_j$. For each object $o_j$ the corresponding portion of the depth image is converted first to a point cloud, then voxelgrids of known-occupied and known-free space centered around the visible object points with a transform $T_j$ mapping the voxelgrid to the workspace coordinates.

For each object $o_j$ we use the Plausible Shape Sampling Network (PSSNet) [5] $f$ to generate possible shape completions. PSSNet is structured as a variational autoencoder. An encoder $f_{enc}$ maps the known-free and known-occupied voxelgrids to a mean and variance in latent space. A latent vector $\psi$ can be sampled and passed to the decoder $f_{dec}$, which outputs a probability of occupancy for each voxel. Thresholding (e.g. $p > 0.5$ for each voxel) yields a completed shape.

An object $o_j$ that is representable by $f$ can be stored compactly as $\psi_j$ such that $f_{dec}(\psi_j) = o_j$. The transform $T_j$ maps the completed shape into the workspace frame. A world is composed of static objects $\{o_1, o_2, ...\} \in \mathcal{O}$. A particle $\phi$ stores a specific world as a sequence of latent-space vectors $\{\psi_1, \psi_2, ...\}$. We sample worlds conditioned on only the depth-image observation by independently sampling latent vectors of objects. The initial belief is a set of particles $\{\phi_1, \phi_2, ...\} \in \Phi$ generated from the information from the depth camera before any robot motion.

## 4.2 Projecting a single object

Sampling particles using only camera information may yield worlds that are inconsistent with the robot contact information. For example, PSSNet may predict objects that extend far into occluded space that intersect regions the robot has moved through. Alternatively, PSSNet may predict objects that do not extend into occluded space, and so the robot may observe contact with no object to explain the collision. Predicting shapes from vision and robot contact in a single pass would require a dataset specific to each robot and a specific set of motions.

To resolve these inconsistencies, sampled particles are projected onto the constraints in the latent space of the shape completion network, shown in Fig. 2. For sample $i$ of object $j$, $\psi_j^i$ induces a workspace occupancy. Our constraints lie in the workspace, but we wish to project the latent space vector. Therefore, the projection is accomplished by optimizing a loss via gradient updates on $\psi_j^i$ while holding $f_{dec}$ fixed, mirroring the process of training a neural network but optimizing the input instead of the network weights. Consider the unthresholded voxelgrid with values between 0 and 1 produced by the decoder: $f_{dec}(\psi_j^i) = W_j^i$.

We optimize the loss: $\mathcal{L}_{all} = \mathcal{L}_{free} + \mathcal{L}_{occ} + \mathcal{L}_{prior}$

The first term $\mathcal{L}_{free}$ penalizes all voxels predicted above a threshold $\delta$ that are known to be free.

$$\mathcal{L}_{free} = \sum_{x,y,z} \begin{cases} \max(W_j^i(x, y, z) - \delta, 0) & \mathcal{W}_{known\_free}(x, y, z) = 1 \\ 0 & \text{otherwise} \end{cases} \tag{3}$$

The second term $\mathcal{L}_{occ}$ penalizes unexplained contact. Each contact $q_{contact}$ must be caused by some object s.t. $\mathcal{R}(q_{contact}) \cap \mathcal{W}_{occ} \neq \emptyset$, however it is not obvious which object is responsible for the contact, or which voxel of the object contacted the robot. During optimization we consider a specific assignment of $Q_{contact}^j$ to object $o_j$. We define the assignment process in Section 4.3. Because a single occupied voxel is enough to explain a contact, at each iteration the loss is optimized based on

the maximum prediction of occupancy overlapping with the collision hypothesis set.

$$\mathcal{L}_{occ} = \sum_{q \in Q^j_{contact}} 1 - \max \left( \mathcal{R}(q) \cdot W^i_j \right) \tag{4}$$

The final term $\mathcal{L}_{prior}$ penalizes deviation of $\psi$ from the original distribution predicted by the encoder. Without this constraint, $\psi$ can deviate arbitrarily, losing all dependence on the depth image and even leaving the training domain of $f_{dec}$. This would produce completions that no longer look like objects. $\mathcal{L}_{prior}$ is weighted by $\alpha$ to maintain a similar magnitude of gradients to $\mathcal{L}_{free}$ and $\mathcal{L}_{occ}$.

$$\mathcal{L}_{prior} = -\alpha \log \left( P(\psi^i_j | f_{enc}(Im)) \right) \tag{5}$$

Sampling the occupancy for a specific object $o_j$ given $Im, Q_{free}$ and $Q^j_{contact}$ is thus accomplished by sampling a $\psi_j$ and optimizing until the constraints are satisfied. Projection can fail if an iteration limit, set to 100 steps, is reached without satisfying the constraints. For practical efficiency this failure can sometimes be detected early when gradient updates no longer change the loss and the constraints are not satisfied. We use Adam [37] for optimization with a learning rate of 0.01.

### 4.3 Multi-object completion

CLASP stores an assignment of each $q_{contact}$ to a particular object $o_j$ for each full-scene particle $i$. When a measurement contains a new contact $q_{contact}$, it is assigned to a specific object for each sampled particle $i$ as follows. First, the output of our shape completer is a finite-sized voxelgrid, typically smaller than the full scene. The new $q_{contact}$ cannot be assigned to any object $j$ where $\mathcal{R}(q_{contact})$ lies entirely outside the output region of the decoder $f_{dec}(\psi_j)$. Next, for each remaining $j$, a projection is attempted for each $\psi^i_j$ to satisfy the new $q_{contact}$. If all attempts fail, we assume this new $q_{contact}$ was not caused by object $j$. For each full-scene particle $i$, a specific assignment of $q_{contact}$ is randomly and uniformly selected from the remaining objects $j$ that could possibly explain the contact.

## 5 Experiments

We evaluated scenarios of different objects to determine if CLASP improves the estimate of the scene using robot contact information. We tested ablations of CLASP to evaluate the importance of the latent prior and constraint satisfaction. We also tested alternative approaches to CLASP that did not rely on projection. Finally, we compared CLASP on two different network architectures and trained on multiple datasets. We trained separate instances of PSSNet [5] on Axis-Aligned Boxes (AAB), YCB [38], and ShapeNet mugs [39] (training details in Section A.1).

### 5.1 Robot Contacts

**Simulation:** To generate contact measurements we moved the right arm of a robot composed of two Kuka iiwa arms with Robotiq 3-finger grippers. We generated scenes by manually placing simulated objects from AAB, YCB, or ShapeNet on a virtual table at about camera height. The known voxels were passed to our trained PSSNet to generate a set of possible completions.

We generated robot motions to gather information by moving near and sometimes contacting the objects using the procedure described in the appendix A.2. The first motion typically sweeps known free space rather than making contact. The second or third motion intentionally makes contact with the object.

**Live Robot:** The physical kinematics of our robot matched the simulated robot. A Kinect depth camera mounted at the "head" position generated the RGBD images. A calibrated motion capture system provided transforms between the Kinect and robot frames. We segmented the RGB image using the CSAIL semantic-segmentation-pytorch library [40] which we retrained on YCB objects. Each segmentation was converted into a voxelgrid and fed to PSSNet as in simulation to generate sample worlds. The same procedure was used to generate robot motions as in simulation.

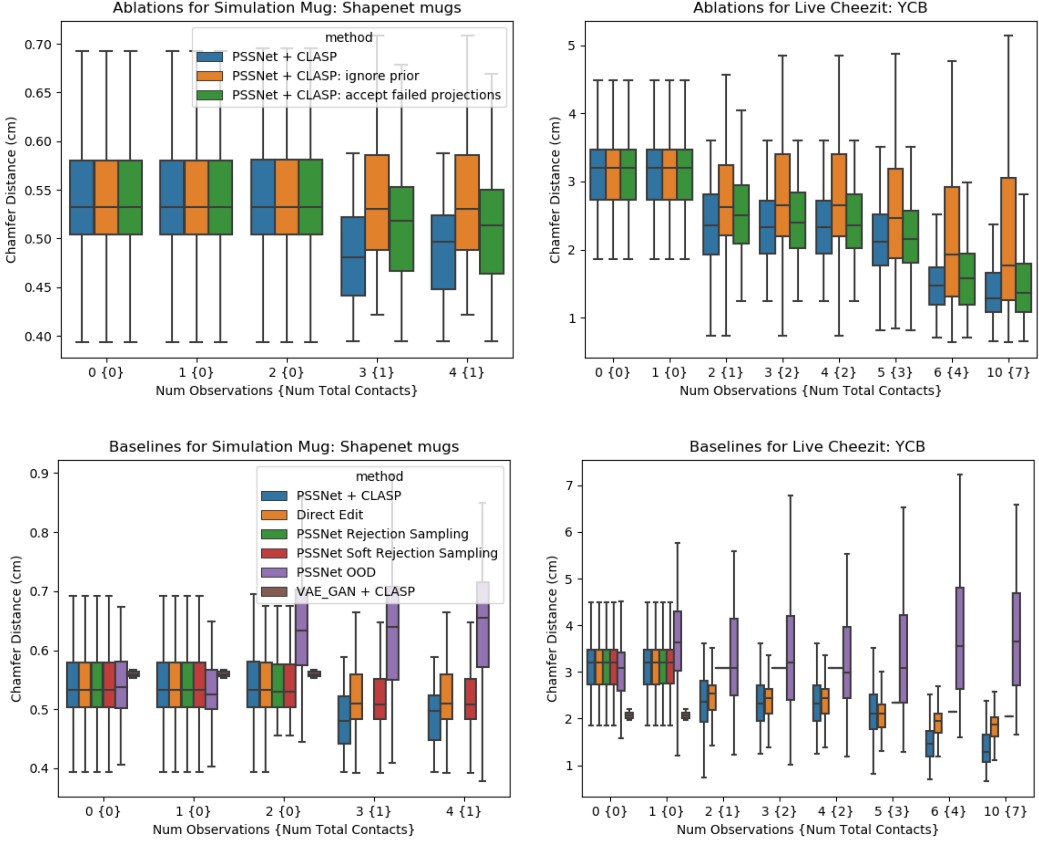

Figure 3: Boxplots showing the Chamfer Distance from sampled particles to ground truth. The mean, middle quartiles (boxed colored region), and outer quartiles excluding outliers are shown. Rejection Sampling and VAE_GAN occasionally produced no valid shapes, in which case no box is displayed.

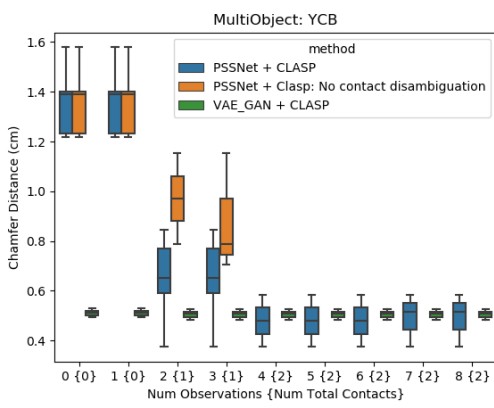

Figure 4: Boxplot results for the multiobject scene. PSSNET + CLASP: NO CONTACT DIS-AMBIGUATION fails to project any samples for observations 4 and beyond, so there is not corresponding box.

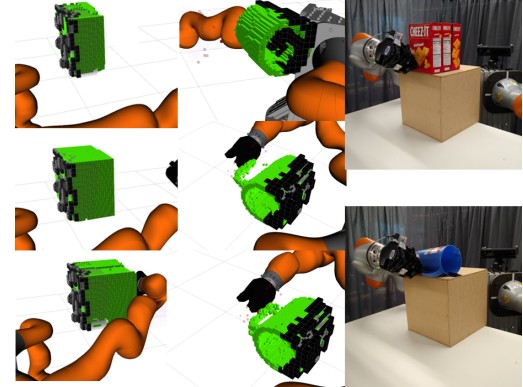

Figure 5: The Deep Cheezit (left), Mug (Middle), Live Cheezit and Live Pitcher (Right) scenes. The occupied (black) and known free (not shown) voxels from vision with contact (transparent red) and robot free (not shown) voxels are all used by CLASP to generate completed shapes (green).

On the live robot, contact was determined at each configuration by checking if the measured external torque exceeded a threshold of $2Nm$ per joint. This threshold was large enough to avoid generating false positives while remaining sensitive enough to detect contact with secured objects.

## 5.2 Scenes

We tested four scenes in simulation and two on the live robot. Each scene consisted of a single object secured to the table in front of the robot. We also tested a scene of multiple YCB objects (Fig. 6). Contacts occurred with occluded sections of the objects, with examples shown in Fig. 5. In both simulation and the live robot, table occupancy was not considered when evaluating the quality of the completions.

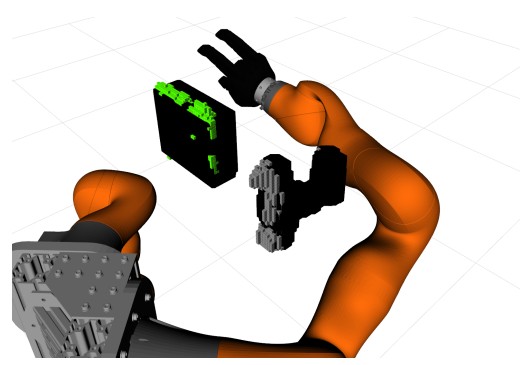

Figure 6: The multiobject scene with the YCB Cheezit box and Drill on a table (not shown) just before the first contact.

**Simulated Scenes:** The first pair of scenarios used a single YCB Cheezit box (Shallow) and a stack of three Cheezit boxes (Deep). These setups generated similar depth images but different ground truth shapes. Both used networks trained on the AAB dataset. The Simulated Pitcher from YCB was positioned with the handle occluded from view and used networks trained on the full YCB dataset. The Simulated Mug from ShapeNet also had the handle occluded from view and used networks trained on all mugs in ShapeNet. The handles on these objects were localized through contact.

**Live Scenes:** The Live Cheezit also consisted of a stack of three boxes, and again the Live YCB Pitcher had the handle occluded. Both scenes used networks trained on the full YCB dataset. The Cheezit boxes were attached together and the pitcher was taped to prevent motion during contact. Simulated objects were manually aligned to the live scene to approximate the ground truth of live objects and were used for evaluation.

## 5.3 Baselines

We compared our proposed method to the following alternatives. DIRECT EDIT directly adds or removes voxels to satisfy the contact information. REJECTION SAMPLING samples latent space vectors from the distribution predicted by the encoder, then decoded these into 3D objects and rejected any samples not satisfying the contact or free space constraints. SOFT REJECTION SAMPLING selects and then Directly Edits the least violating samples in the cases where all samples are rejected. For OOD (Direct Out-Of-Distribution Prediction), we combined the visual known free and occupied voxels with the contact known free and occupied voxels as input to PSSNet. While other methods did not have access to the true contact point, we allowed this method this advantage and added the true contact point directly to the known occupied voxels from the depth image. Finally, while other approaches used the PSSNet shape completion network, we tested using a VAE_GAN [17] network. This network tends to produce better average but less diverse samples. Section A.3 describes these baselines in more detail.

We also tested ablations of our method. CLASP: IGNORE PRIOR tested removing the loss term $\mathcal{L}_{prior}$. CLASP: ACCEPT FAILED PROJECTIONS tested accepting all projections, even those that do not satisfy the contact constraints. To test our contact assignment in the multi-object case (Section 4.3), CLASP: NO CONTACT DISAMBIGUATION determined if a projection of latent $\psi_j$ could satisfy each new $q_{contact}$ as in CLASP, then assigned each $q_{contact}$ to all feasible objects $j$. This resulted in scenes explaining a single $q_{contact}$ with multiple objects.

100 particles were sampled in each method, with the threshold of $\mathcal{L}_{free}$ set at $\delta = 0.4$ and the weighting of $\mathcal{L}_{prior}$ set at $\alpha = 0.01$.

### 5.4 Results

Single-object scenes were tested on all baselines and ablations using PSSNet trained with the appropriate dataset. Fig. 3 compares the Chamfer Distance (CD) [41] of each accepted sample to the ground truth for selected scenes. Plots for all scenes are shown in Section A.3. We consider a different analysis in Section A.4. We find that REJECTION SAMPLING often performs well during the first few observations with zero or one contact. However, REJECTION SAMPLING soon fails to return any valid samples with two or more contacts. We see that OOD produces completions that are typically much worse than the original completions from only vision. The initial estimate (observation 0) from VAE_GAN are hit-or-miss. All networks saw the YCB Pitcher during training, and VAE_GAN recalls the pitcher more accurately than PSSNet during testing to the point where contacts are unnecessary. However, the recall of VAE_GAN in the other ambiguous scenarios is worse than PSSNet and projection to the contact constraints often fails, leaving no sampled shapes. The results justify our choice of PSSNet over VAE_GAN for CLASP, as VAE_GAN is unable to sufficiently adjust to the contact information.

Considering ablations of CLASP, ACCEPT FAILED PROJECTIONS performed as well initially (when no projections fail) and significantly worse as the number of observations increases. Ignoring the latent prior during projection also performs worse and occasionally produces shapes that qualitatively look less like objects compared to completions from other methods.

Across all scenes, CLASP performed similarly to the best of all other methods with 0 or 1 contacts and the best with multiple contacts. CLASP successfully used the robot contact information in all scenarios to reduce the CD between the predicted and ground truth shapes in all scenarios. Robot measurements with a contact typically caused a larger reduction in CD than measurements with only freespace information. Numerically, the CD reduced most in the Cheezit scenarios, with a reduction of the mean from $0.5cm$ to $0.1cm$ for the Shallow and from $0.14cm$ to $0.08cm$ for the Deep. The CD reduction in the pitcher and mug scenarios was significantly smaller, as the general shape of the pitcher and mug could be predicted from the image. The prediction of the occluded handle was improved with contact. The trend of improvement in the live scenes matched the simulation. However, the numeric error of the live scenes was much larger, perhaps caused by imperfect transfer of the learned shape completer from training in simulation to prediction on live Kinect data as well as imperfect alignment of the robot frame to the Kinect frame.

In the multi-object scene (Fig. 4) the VAE_GAN method achieves a better completion from the RGBD data, but our proposed method produces better samples after 2 contacts. Our proposed contact assignment (Section 4.3) outperforms naively satisfying the contacts whenever possible.

## 6  Discussion and Conclusion

While we model CLASP using a particle filter and would like to have the Bayesian estimate of the scene given all observations, we acknowledge many non-Bayesian approximations. Particle filters approximate Bayes filters, but the 100 particles we sample may not be a sufficient coverage of the latent shape space. CLASP projects samples, which does not preserve Bayesian estimates.

We demonstrated a method for estimating shape completions initialized with purely RGBD visual data, then updated from observations of a robot arm moving through unknown regions and sensing contact. We stored the belief of the scene as a particle filter of latent vectors from a shape completion network and used CLASP to enforce shape consistency with the robot observations. Most importantly, we showed that CLASP improves the estimate of object shape using these contact observations. Our results further showed that CLASP performs better than ablations of CLASP and alternative methods. We hope CLASP will be used within a larger robotics framework where reasoning over environment uncertainty based on shape priors aids in accomplishing larger goals.

**Acknowledgements:**  This work was supported in part the Toyota Research Institute and by NSF grant IIS-1750489 and ONR grant N00014-21-1-2118. This article solely reflects the opinions of its authors.

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
