# OpenReview forum: "CLASP: Constrained Latent Shape Projection for Refining Object Shape from Robot Contact"
_robot-learning.org/CoRL/2021/Conference — CoRL2021 Poster_

### Official Review · Reviewer_wjGX · 2021-07-21

**Originality:** Good
**Technical Quality:** Very Good
**Clarity Of Presentation:** Very Good
**Impact:** 3

**Recommendation:**

Weak Accept: I recommend accepting the paper, but will not argue for my recommendation if the majority of other reviewers have a different opinion.

**Summary:**

The paper presents Constrained Latent Shape Projection (CLASP), a method for refining object shape (originally inferred from vision) with information from robot contact/tactile information. The overall framework involves a sensor fusion which is based on a particle filter, and using CLASP --which is a projection of the particles/samples onto contact constraints in the latent space-- as the measurement update. The projection technique itself is quite interesting: it's similar to the training process of the neural network (via backpropagation algorithm), but instead of optimizing the weights, it optimizes the inputs. In the experimental section, ablation studies as well as comparison with other method such as VAE-GAN are performed on both simulation and a real robot.

**Issues:**

Please see the points mentioned in the "Weaknesses" section above as well as a few additional suggestions below, and please try to address them. Thanks!

Additional suggestion(s):
(1) In the experimental results/video, it would be nicer if the authors could highlight regions of the point clouds that were cut/chamfered due to the additional contact information, to get a better sense/reasoning of how the trained model behaves.

**Reviewer Expertise:**

Good: General knowledge of the area

**Strengths And Weaknesses:**

Strengths:
(1) The constraint latent space projection technique is quite interesting: it's similar to the training process of a neural network (via backpropagation algorithm), but instead of optimizing the weights, it optimizes the inputs.
(2) In the experimental section, ablation studies as well as comparison with other method such as VAE-GAN are performed on both simulation and a real robot.
(3) Source code is provided in the supplementary materials.

Weaknesses:
(1) A missing reference: S. Luo, W. Mou, K. Althoefer and H. Liu, "Localizing the object contact through matching tactile features with visual map," 2015 IEEE International Conference on Robotics and Automation (ICRA), 2015, pp. 3903-3908, doi: 10.1109/ICRA.2015.7139743.
(2) (a) Figure 2 is not very clear: what are the pink and blue rectangles in between the "sample" arrow and the "f_{dec}" trapezoid? Maybe worth checking if other readers understand this figure well, including each components drawn in it. Does each component of the figure provide a useful information?
(b) Figure 1: bottom right figure (third row, vision + all contacts) is missing?

**Summary Of Recommendation:**

While the technique presented in the paper is interesting, the way of communicating it to the reader (e.g. via the figures) may need to be improved.

---

> ### Author Response · Authors · 2021-08-23
> **Response to wjGX**
>
> We have added the reference.
>
> Figure 1 is correct as is. The bottom row is intentionally left blank, illustrating CLASP successfully determining that the objects in the bottom row are not feasible.
>
> In Figure 2, the rectangles represent intermediate states of the PSSNet neural network. The details of PSSNet are available in the referenced paper. Based on your suggestion we have labeled the mean and logvariance rectangle in the figure, referenced PSSNet in the figure caption, removed an intermediate rectangle that was not necessary, and explicitly explained $f_{enc}$ and $f_{dec}$ in the method section. End encoder, latent prior, latent sample $\psi$, and decoder are necessary for the understanding of the method. We removed the unnecessary box to simplify the figure.
>
> Thanks for your suggestions

---

### Official Review · Reviewer_URGm · 2021-07-22

**Originality:** Good
**Technical Quality:** Good
**Clarity Of Presentation:** Very Good
**Impact:** 3

**Recommendation:**

Weak Accept: I recommend accepting the paper, but will not argue for my recommendation if the majority of other reviewers have a different opinion.

**Summary:**

The paper presents a method to refine single-view shape completion predictions through a combination of free-space carving and robot collisions. The approach iteratively updates a believe over object shapes using a particle filter in the latent space of a shape completion network. It is demonstrated that the approach can improve the initial shape predictions, both in simulation and in a robotic experiment, for objects that are quite close to the objects used during training.

**Issues:**

As mentioned before, the paper is well written. Some of the general mathematical descriptions (e.g. Section 3) are not really adding anything to the text, but this is a matter of taste.

**Reviewer Expertise:**

Very good: Comprehensive knowledge of the area

**Strengths And Weaknesses:**

Strength:

- The approach takes into account the ambiguities of single-view object reconstructions by refining a distribution over possible shapes
- The problem, i.e. how to update the believe of previously predicted 3D geometry when encountering robot collisions, is challenging and relevant
- It is shown that by reintroducing constraints such as very similar objects with simple shapes during training and testing, collision detection and free-space carving can improve results
- Both, the paper and the video contain a clear presentation of the chosen concepts
- The paper discusses problems and limitations of the presented approach that are related to the underconstrained nature of the problem.
- Limited results can still be important to initiate further research.

Weaknesses:

- The tackled problem is severely underconstrained and we can therefore not expect good results:
1.) Shape completion ambiguities through partial views
2.) No knowledge of the contact position
3.) No knowledge on the environment
4.) Static scene assumption while dealing with collisions
- The problem can only be made tractable by reintroducing a lot of constraints that severely limit the general applicability of the approach:
1.) Objects during training and testing are very similar
2.) Only simple shapes are considered
3.) The objects are fixed to the ground so that they do not move
4.) Multi-object setup is basically a wider version of a single object (stacked boxes)
- The rejection sampling sometimes works better until an additional contact makes all shapes implausible. I would have liked to see some sort of soft rejection sampling as a baseline since the estimated contact positions are only broad approximations.
- The baselines are mostly ablations of the presented method. Alternative baselines are not considered. E.g. since the shape completion networks anyways do not generalize over multiple categories, we could also use intra-class shape retrieval / parametric shape estimation / categorical pose estimation. Collisions at the back of the object could still be used to (softly) constrain the selection of the right pose/shape.
- The applicability of the approach in a downstream-task is not shown. At this accuracy, I could only imagine motion planning, but then you could also make the planner react to collisions and adapt its plan, directly. Or by just moving the camera / having two cameras to remove ambiguities.
- It is unclear how much CLASP helps the (often better) VAE-GAN results, I can only observe a gain in the simulation results of cheezeit (shallow) over number of contacts

**Summary Of Recommendation:**

I am leaning towards weak accept, mainly, because the paper and video are well crafted, but I could also understand if other reviewers lean towards rejection because the practicality is questionable. The method of iteratively refining latent codes is not new as also noted in the paper [30]. The authors try to use even less information to locate contact points (just collision information). I believe that to make the problem tractable and to achieve the accuracy needed in robotic applications we not only need better shape completion methods but also touch-sensitive skin. Trying to circumvent this via constraint optimization seems a bit hopeless but the general method might become interesting when these components are available.

---

> ### Author Response · Authors · 2021-08-23
> **Response to URGm**
>
> Based on this review we have added:
> The Soft Rejection Baseline
> A “Direct Edit” baseline
>
> This review repeats the constraints mentioned in the problem statement of this work. Such constraints do limit the general applicability of the proposed method, and we welcome any work that removes these constraints. In particular, we welcome improved shape completion methods and methods that can track partial objects thus removing the fixed-object constraint. Both improvements would expand this work as well as much other work on shape completion.
>
> “Multi-object setup is basically a wider version of a single object”
> This is incorrect. We believe the confusion was due to figure 4 referencing the multi-object setup, while the adjacent figure 5 shows single object scenarios.
>
> This review also suggests a set of other methods, such as intra-class shape retrieval, with contact used to constrain the pose. While we agree such a method is possible, we believe such a comparison is beyond the scope of this paper. There is an open question as to whether neural networks are a good approach for shape completion, yet neural network approaches have never-the-less become incredibly popular recently and for this applications shown strong results. This paper does not weigh in on this larger debate. Furthermore, while such a method is possible, We are not aware of a specific implemented method that combines this shape retrieval from both vision and contact. Further baselines could bolster the results of this paper, yet this specific comparison would add a substantial amount of work.
>
> This review suggests shape completion could be improved by adding additional cameras, which while true, may not be possible given the constraints of the problem and real robots. In a lab or factory it may be possible to add cameras at additional locations. Our lab experiments were proxies for scenarios where field of view is limited due to limited robot motion or tight spaces such as a cabinet. Indeed, much work on shape completion assumes objects are not visible from all angles.
>
>
> We welcome improvements to robotic touch-sensitive skin. However, we expect it will take a long time for such skin to become widespread, thus we believe work such as this aiding contact representations without skin is valuable. As noted in this review, the proposed method is still applicable to robots with skin.

---

> > ### Comment · Reviewer_URGm · 2021-08-24
> > **response**
> >
> > I appreciate the Soft Rejection Baseline.
> >
> > > “Multi-object setup is basically a wider version of a single object” This is incorrect. We believe the confusion was due to figure 4 referencing the multi-object setup, while the adjacent figure 5 shows single object scenarios.
> >
> > It is still unclear to me how your multi-object setup looks like, if this does not refer to the 3 YCB-V Cheeze-It boxes stacked together (that is what I mean by "a wider version of a single object"). You could just show an image in the appendix and describe it clearly, otherwise it seems like you don't really want to specify it.
> >
> > > Furthermore, while such a method is possible, We are not aware of a specific implemented method that combines this shape retrieval from both vision and contact. Further baselines could bolster the results of this paper, yet this specific comparison would add a substantial amount of work.
> >
> > I think the burden of limited results is the demand for additional baselines. In that case the added complexity from contacts needs to be justified by favorable results against *multiple* vision-only approaches from the shelf, otherwise it will not be adopted. E.g. your VAE-GAN baseline is a good start and gives us a lot of insights. We don't want to end up with papers where a simple baseline can beat more complex approaches:
> > https://openaccess.thecvf.com/content_CVPR_2019/html/Tatarchenko_What_Do_Single-View_3D_Reconstruction_Networks_Learn_CVPR_2019_paper.html

---

> > > ### Author Response · Authors · 2021-08-25
> > > **Added Multi-object Figure**
> > >
> > > Thank you for pointing out the multi-object setup was not sufficiently specified. We have updated the main paper, adding figure 6 showing the multi-object scene, consisting of the YCB cheezit box and the YCB drill.

---

> > > > ### Comment · Reviewer_URGm · 2021-09-03
> > > > **Limited but interesting**
> > > >
> > > > Overall, the authors have slightly improved the paper with the new baselines. I still find the setting severely under-determined and think that this way of approaching the problem quickly hits a wall. But tackling hard problems and then achieving limited results shouldn't be the only reason to reject a paper, especially when it is well presented and if there are open reviews that allow for a neutral perspective. My score remains unchanged.

---

### Official Review · Reviewer_82yA · 2021-08-16

**Originality:** Good
**Technical Quality:** Good
**Clarity Of Presentation:** Very Good
**Impact:** 3

**Recommendation:**

Weak Reject: I recommend rejecting the paper, but will not argue for my recommendation if the majority of other reviewers have a different opinion.

**Summary:**

The authors want to learn object shape from sensors on the robot. As a starting point, they apply a baseline method that uses solely RGBD data, using the prior work of Plausible Shape Sampling Network (PSSNet). This prior work first converts the RGBD images into voxel grids that are known-free and known-occupied. The PSSNet is then a VAE that encodes + decodes into a latent space. Latent vectors $\psi$ can then be sampled, decoded into occupancy probability estimates, and aggregated into object shapes.

This work aims to augment PSSNet with robot contact information. The initial belief of objects is represented as a particle filter of several samplings $\psi$ from the prior. As the robot moves in the environment, the contact sensors lead to contact constraints (either from free space or detected contacts). Given an already-trained PSSNet, the proposed method (CLASP) optimizes over $\psi$ to get new latents that are consistent with the contact constraints. The loss includes a term that encourages $\psi$ to stay close to the prior. Thus over the course of interaction, the potential object shapes become more constrained. This is then compared against other ways of integrating the contact information (such as rejection sampling).

**Issues:**

I would appreciate stronger comparisons to simple methods of using contact info to constrain the prior and would be willing to improve the rating if this were done.

**Reviewer Expertise:**

Good: General knowledge of the area

**Strengths And Weaknesses:**

I am not too familiar with shape completion, but the method seems reasonable. I did not understand what the OOD baseline meant - it is clear that RGBD data is augmented with contact information in some way, but it's not clear what this means. My interpretation was that contact information was propagated to the PSSNet in some way, and predictions from PSSNet were used directly, rather than doing a live constraint update like CLASP does, but if this is how that baseline works, the details are fairly sparse.

To me the main weakness is on the baselines. The main baseline used is rejection sampling, which the authors note is fairly competitive until they hit a point where the rejection sampler returns 0 valid object configurations. Given that rejection sampling was fairly effective, this makes me wonder how well the method does if the rejection sampling selects the object that violates the *minimum* number of constraints rather than requiring 0 constraint violation. It seems unfair to exclude comparisons to rejection sampling at all in the higher # of contacts setting. I am also curious about more complex MCMC algorithms in general for baselines.

The other main weakness of the method is that it is fairly reliant on the shape prior. As the authors noted, when they tried using a VAE-GAN for the shape prior, they found that average samples were more accurate, but samples were also much less diverse, and this harmed the contact info update. Thus, the effectiveness of CLASP seems tightly coupled to properties of the learned encoder / decoder...but this isn't really the focus of the paper, the paper is focusing on how to augment the shape prior, so I am not considering this for the review.

**Summary Of Recommendation:**

The method is very reasonable but I am unsure of its effectiveness.

---

> ### Author Response · Authors · 2021-08-23
> **Response to Reviewer 82yA**
>
> Based on your comments We have added two additional baselines as well as lengthened the explanation of the OOD baseline.
> We have added the “Direct Edit” baseline, and based on your suggestion We have added a “Soft Rejection Sampling”. As you suggested, when all samples are rejected we instead choose the samples that have the minimum number of constraint violations. We then improve upon this method by applying “Direct Edit” to these remaining samples, ensuring each sample satisfies all constraints. Empirically, this performs better than the exact baseline you suggested. This “Soft Rejection Sampling” performs worse than simply using the last valid pure rejection sampling approach.
>
> More baselines can always bolster the arguments of a paper. Please consider that this paper now has 7 ablations and baselines, whereas many similar papers have 2 to 4.
>
> You are correct that the method is reliant on the shape prior. We are fundamentally attempting to predict shape from a severely under-constrained space, thus the prior will play a large role in the sampled shapes. The intended takeaway of the VAE-GAN results are precisely as you have described and highlight the value of the diversity of using PSSNet. We do not see the reliance on a shape prior as a weakness, but instead as a crucial component of inferring shape from limited data.

---

> > ### Comment · Reviewer_82yA · 2021-08-26
> > **Re: Response**
> >
> > Thanks for adding the Soft Rejection Sampling baseline.
> >
> > For the shape prior, the reason I see this as a weakness is because neural net shape priors are not necessarily trained to have high diversity in their samples. My concern is that as neural net shape priors improve, they may not maintain the diversity needed for methods like these to work - I would argue that the field tends to reward precise predictions of incomplete data, rather than good coverage of potential completions. So the importance of this work is conditional on the way in which shape priors improve over time.

---

### Author Response · Authors · 2021-08-23
**Response to all reviews**

Thank you for the reviews. The reviewers have stated the paper is “well written”, the video is “well crafted”, the “technique presented is interesting”, and “the method is very reasonable”. We particularly appreciate that Reviewer URGm lists “This paper discusses problems and limitations…” as a strength, as we believe discussions around the limitations is a crucial yet undervalued aspect of robotics papers.

The most common criticism from the Reviews were suggestions for additional baselines. Based on this feedback, we have added a “Direct Edit” and a “Soft Rejection Sampling” baseline.Additional baselines can always bolster the arguments of a paper. However, please consider that this paper now has 5 baselines and 2 ablations. Similar papers have 2 (Multi-Fingered Active Grasp Learning, Lu et al. IROS 2020), 3 (3D shape Perception from Monocular Vision, Touch, and Shape Priors, Wang et al. IROS 2018), and 4 (Amodal 3D Reconstruction for Robotic Manipulation via Stability and Connectivity, Agnew et al. CoRL 2020) total baselines + ablations.

Reviewers suggested 5 additional references, which we added to the Related Work.

Reviewer xjGX had specific suggestions for Figures 1 and 2 which we implemented or responded to.

Reviewer URGm had concerns that the problem formulation was both too challenging and too limiting. We agree the problem is challenging and the current method requires restrictions we would ultimately like to remove. We also believe predicting the shape is a valuable step towards enabling robots to operate in unstructured environments. We respond to high level comments from the reviewer below
“...and we can therefore not expect good results”.  We agree we cannot expect results as good as are often presented in shape completion papers trained on single objects, yet robots can accomplish many tasks without reconstructing picture-perfect objects. We believe my results show the quality of completions we can expect, and they show a clear improvement by using contact information, which was the goal of this method.

“I could only imagine motion planning, but then you could also make the planner react to collisions and adapt its plan”.  We do indeed make a planner that reacts to collisions and adapts its plan (Appendix A.2). Adapting a plan requires somehow considering the contact information, which we accomplish by updating a belief of the world occupancy using CLASP. This work enables robots to replan using contact information.

“I believe that to make the problem tractable…[we need] touch-sensitive skin”. We welcome any improvements to robotic skin, however we expect robots without skin will be common for a long time. This work can aid robots both with and without skin.


I respond to other comments in the replies to reviewers below. Again, thank you for the detailed reviews.

---

### Meta-Review · Area_Chair_6owX · 2021-08-17

**Recommendation:** Accept (Poster)
**Confidence:** 4

**Metareview:**

This paper presents an extension to the PSSNet diverse object shape reconstruction work to incorporate reconstruction from contact sensing in addition to vision.

In the updated version the authors provide a comprehensive set of baseline comparisons and do a better job of situating it in the broader reconstruction literature.

As such I think this would make a nice addition to the conference.

I would recommend the authors do another pass through the comments of the reviewers to make sure everything is correctly addressed. I also think I suggested the wrong reference from Lu in my previous meta-review, as I intended the authors to include a discussion and reference of their reconstruction work ("Learning Continuous 3D Reconstructions for Geometrically Aware Grasping" https://ieeexplore.ieee.org/abstract/document/9196981)

---

> ### Author Response · Authors · 2021-08-23
> **Response to Area Chair**
>
> We have added the “Soft Rejection Sampling” baseline as well as a “Direct Edit” baseline. We have made all changes suggested to improve the clarity of the presentation.
>
> We have expanded the related work section, adding the 4 references listed.
>
> We defend the importance of this work in text and in the other rebuttals.

---

### Decision · Program_Chairs · 2021-09-13

**Decision:**

Accept (Poster)

**Comment:**

This paper presents an extension to the PSSNet diverse object shape reconstruction work to incorporate reconstruction from contact sensing in addition to vision.

In the updated version the authors provide a comprehensive set of baseline comparisons and do a better job of situating it in the broader reconstruction literature.

As such I think this would make a nice addition to the conference.

I would recommend the authors do another pass through the comments of the reviewers to make sure everything is correctly addressed. I also think I suggested the wrong reference from Lu in my previous meta-review, as I intended the authors to include a discussion and reference of their reconstruction work ("Learning Continuous 3D Reconstructions for Geometrically Aware Grasping" https://ieeexplore.ieee.org/abstract/document/9196981)